# Identification of Novel Antimicrobial Resistance Genes Using Machine Learning, Homology Modeling, and Molecular Docking

**DOI:** 10.3390/microorganisms10112102

**Published:** 2022-10-23

**Authors:** Janak Sunuwar, Rajeev K. Azad

**Affiliations:** 1Department of Biological Sciences and BioDiscovery Institute, University of North Texas, Denton, TX 76203, USA; 2Department of Mathematics, University of North Texas, Denton, TX 76203, USA

**Keywords:** machine learning, antimicrobial resistance, homology modeling, molecular docking

## Abstract

Antimicrobial resistance (AMR) threatens the healthcare system worldwide with the rise of emerging drug resistant infectious agents. AMR may render the current therapeutics ineffective or diminish their efficacy, and its rapid dissemination can have unmitigated health and socioeconomic consequences. Just like with many other health problems, recent computational advances including developments in machine learning or artificial intelligence hold a prodigious promise in deciphering genetic factors underlying emergence and dissemination of AMR and in aiding development of therapeutics for more efficient AMR solutions. Current machine learning frameworks focus mainly on known AMR genes and are, therefore, prone to missing genes that have not been implicated in resistance yet, including many uncharacterized genes whose functions have not yet been elucidated. Furthermore, new resistance traits may evolve from these genes leading to the rise of superbugs, and therefore, these genes need to be characterized. To infer novel resistance genes, we used complete gene sets of several bacterial strains known to be susceptible or resistant to specific drugs and associated phenotypic information within a machine learning framework that enabled prioritizing genes potentially involved in resistance. Further, homology modeling of proteins encoded by prioritized genes and subsequent molecular docking studies indicated stable interactions between these proteins and the antimicrobials that the strains containing these proteins are known to be resistant to. Our study highlights the capability of a machine learning framework to uncover novel genes that have not yet been implicated in resistance to any antimicrobials and thus could spur further studies targeted at neutralizing AMR.

## 1. Introduction

Frequent incidences of antimicrobial resistance (AMR) in hospital and community settings in recent years call for urgent intervention and new strategies to control this burgeoning health problem. AMR has also been in the news during the ongoing COVID-19 pandemic, with Centers for Disease Control and Prevention (CDC) reporting increased instances of hospital-onset antibiotic resistance among COVID-19 patients [1]. This overburdened an already stretched healthcare system in the pandemic. Globally, the mortality due to AMR is projected to be 10 million per year, with an economic burden of USD 300 billion that could shoot up to USD one trillion by 2050 [2]. The current arsenal of antibiotics to combat infections leaves much to be desired due to the festering AMR problem. Loss of drug efficacy has been exacerbated by the lag in development of new antibiotics since the late 1990s [3,4]. Unfortunately, the reports on antibiotic research and development (until 2017) indicate a lack of incentives, where only about USD $46 million per year in revenue is generated from the sale of antibiotics, while the investment is manifold at USD $1.5 billion [5]. Due to these factors, there has been an innovation gap in the development of antibiotics and as a consequence, alternative therapeutics including various biomolecules and nucleic-acid based alternative systems such as CRISPR-Cas-based antimicrobials, peptide nucleic acids, bacteriophage therapies, antibodies, bacteriocins, and anti-virulence compounds have been used [4,6]. Emergence of new resistant strains via creation and evolution of new resistance genes or acquisition of novel combinations of resistance genes muzzles the efforts to mitigate AMR. Selection allows the genomic landscapes of superbugs to be continually re-shaped by mutation, insertions, deletions, horizontal gene transfer, and genomic rearrangements [7,8].

Considering the current state of affairs in tackling AMR, new strategies to identify and prioritize yet unknown genetic factors responsible for AMR are urgently needed. Here, we leveraged the advances in machine learning or artificial intelligence and high-throughput technologies for genotyping and phenotyping, to uncover novel AMR genes in bacterial strains. Machine learning models have previously been employed to learn complex patterns underlying the association between genotypes and phenotypes, followed by application of the trained models to predict phenotypes given the genotypes. Machine learning has also shown promising outcomes in the investigations of AMR, where the prediction of a strain as susceptible or resistant by a model is often based on genetic and antimicrobial susceptibility testing (AST) data [9,10,11]. Recently, we proposed a machine learning framework to predict genetic factors in bacterial pathogens that are responsible for resistance to specific antibiotics [12]. Most of these studies have focused on known AMR genes and assessed their roles in the resistance to antibiotics other than those they are known to be resistant to [12]. This, however, restricts the capability of machine learning in identifying novel resistance genes that have not yet been implicated in the resistance to any antibiotics. To tap the full potential of machine learning in deciphering resistance genes, we considered the full spectrum of genetic data, that is, the entire set of annotated genes in a genome of interest and used it in conjunction with the respective AST data within an established machine learning framework, in order to predict novel genes underlying resistance to different antibiotics.

The known functions of predicted resistance genes hint at their potential roles in resistance. However, our analysis also uncovered putative resistance genes that are yet to be characterized for their functions (the so called “hypothetical protein” genes). To further validate our predictions, we conducted molecular docking studies, which are frequently used in computer-aided drug design (CADD) to predict interactions between small molecules (ligands) and proteins (receptors) based on the stability of their docking conformations [13,14,15]. The fidelity of the predictions was based on the likelihood of stable interactions between the predicted proteins as receptors and the respective antibiotics as ligands, assessed in terms of their binding affinities. Our pipeline thus aids in prioritizing genes potentially involved in AMR for further downstream analysis in a wet lab setting.

## 2. Materials and Methods

For each bacterial species and specific antibiotic, genotype and phenotype data download, data filtration, hypothetical protein re-nomenclature, training, testing, and validation, optimal ML model determination, identification of novel AMR loci, homology modeling, and molecular docking were performed individually. Details of the data and procedure are provided below.

### 2.1. Bacterial Isolates

The Isolates with genotypic and AST phenotypic data were manually filtered for each bacterial species from the NCBI Pathogen Detection database: https://www.ncbi.nlm.nih.gov/pathogens/ (accessed on 15 May 2020). The AST phenotypes here refer to the antibiogram of the BioSample database, which includes the information on antibiotic susceptibility phenotypic testing, Minimum Inhibitory Concentration (MIC) quantifying the minimum concentration of an antibiotic inhibiting the growth of a bacterium, or disk diffusions. The genotype here refers to all annotated genes of a strain, as in the NCBI Pathogen Detection Database. The isolate data were retrieved only for those strains that have both genotype and AST data available, as all the annotated genes present in the strains were considered as features and the AST phenotypes were considered as “labels” while training a model to predict the phenotypes. To maintain uniform feature names across all samples, RefSeq assembly summary and RefGene were used to download the feature table for each isolate to access all annotated genes for features extraction. The union of genes from all the isolates comprised the full feature set. Then accordingly, a binary gene presence/absence (gene presence coded 1 and gene absence coded 0) was entered in the feature table for each gene from each sample. Similarly, AST phenotypes (resistant/susceptible) for a specific antibiotic were binarized (resistant coded 1 and susceptible 0), and an AMR-AST matrix was thus constructed for each antibiotic–species group combination considered in this study. The final processed matrix consisted of sample accession numbers, features (genes), and AST labels (resistant/susceptible) as target classes, for inputting into various machine learning algorithms prepped for a supervised binary classification task. Note that among the phenotypes of each strain as referenced by the CLSI and EUCAST standard, the “intermediate” (I) has been dealt as “resistant” (R) in this study. Details on sample size, feature size, and labels are provided in Table 1, and all processed datasets have been made available at project GitHub site: https://github.com/Janaksunuwar/AMRprediction_HomologyModeling_Docking. (accessed on 15 May 2020)

### 2.2. Hypothetical Protein Re-Nomenclature

Hypothetical proteins/genes may have different sequences but are all named “hypothetical protein” in the NCBI RefSeq feature table. For our analysis, these sequences needed to be differentiated, with highly dissimilar ones deemed different features in supervised learning. For each bacterial species considered in this study, sequences of all hypothetical proteins from each strain were obtained, and all against all protein BLAST (BLASTp) was performed. A common name was given to hypothetical proteins that were deemed homologs based on the following BLAST criteria: 70% query coverage and 30% identity. These re-nomenclatured hypothetical proteins were reassigned to their respective feature tables to be considered as genetic features for the binary matrix. Note that each hypothetical protein coding gene has its own accession number assigned to it by NCBI; our new nomenclature was not intended to replace the existing standardized nomenclature but was done solely for the purpose of this focused study.

### 2.3. Training, Validation, and Testing

For each bacterial species and antibiotic combination, instead of a single round of split of the dataset for training, validation, and testing, a total of six rounds of split (6-fold stratified cross-validation [12,16], implemented using StratifiedKFold of scikit-learn available at https://scikit-learn.org/stable/ accessed on 15 May 2020) were performed on the dataset to ensure that features were sampled from the entire data for model training and testing, as further detailed below (Section 2.6). Here for each round of split of the data, we refer it as a set and have a total of 6 sets. For each set, the data was partitioned into 6 equal parts, with 5 parts (i.e., 5/6th or 83.33% of the dataset) considered as the training set and the remaining part (1/6th or 16.66%) considered as the held-out test set. Then for each 5/6th training, a nested 10-fold cross-validation and a leave-one-out (Loo) cross-validation were also performed, and the trained model was tested on the held-out dataset. Performance was assessed by averaging the accuracy scores generated in the six rounds of cross-validation. Note that the same split strategy was applied uniformly throughout this study. The workflow illustrating this procedure is shown in Figure 1.

### 2.4. Machine Learning Algorithms

The following machine learning algorithms available at scikit-learn (https://scikit-learn.org/stable/ accessed on 15 May 2020) were used in this study: Logistic Regression (logR), Gaussian Naive Bayes (gNB), Support Vector Machine (SVM), Decision Trees (DT), Random Forest (RF), K-Nearest Neighbors (KNN), Linear Discriminant Analysis (LDA), Multinominal Naive Bayes (mNB), AdaBoost Classifier (ABC), Gradient Boosting Classifier (GBC), ExtraTrees Classifier (ETC), and Bagging Classifier (BC). The default algorithm parameters, along with custom optimal parameters, are provided with the source code of the ML framework at the project’s GitHub site (see Data Availability section below), and in Appendix A.

### 2.5. Determination of Optimal Machine Learning Framework

For each bacterial species and antibiotic combination, machine learning algorithms were trained and tested on whole genome genotype (all genes) and AST phenotype data. The performance metrics, namely, recall, precision, F1 score, area under the receiver operating characteristic (AU ROC), and area under the precision-recall curve (AUPR) were computed for both training and test datasets (referred to as training performance and test performance respectively) under 6-fold, nested 10-fold, and leave-one-out (Loo) cross-validations. The performance was assessed as the average over *n*-round accuracies in an *n*-fold cross-validation. The model that yielded the maximal overall accuracy (F1 score) was selected as the optimal model, and following this, the most important features (genes) were enlisted for further analysis as described below.

**Figure 1 microorganisms-10-02102-f001:**
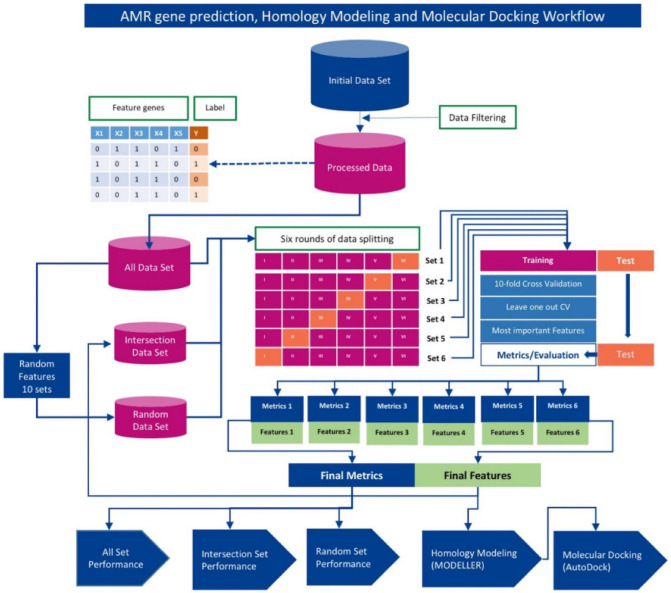
Schematic diagram of AMR trait and gene prediction pipeline that integrates machine learning, homology modeling, and molecular docking. The input data matrix contained gene presence/absence information (X_i_ for ith gene; 1 denotes presence and 0 denotes absence) and phenotype information (Y; 1 denotes resistant and 0 denotes susceptible) for each strain (each row in the matrix). Assessment of machine learning (ML) algorithms was conducted using (i) All Set: entire gene dataset, (ii) Intersection Set: genes deemed important for discrimination and appeared consistently in all 6 rounds of cross-validation, and (iii) Random Set: randomly sampled genes (same number of genes as in Intersection set). Six-fold cross-validation was performed yielding performance of machine learning algorithms on respective data sets (i–iii). The Intersection Set genes were found to be yielding overall optimal performance and were therefore subjected to homology modeling and molecular docking analyses to assess whether protein-products of these genes form stable conformations with corresponding ligands (antibiotics) in molecular simulations.

### 2.6. Identification of Novel AMR Genes/Loci

First, the performance of machine learning algorithms was assessed using all genes in each round of training, validation, and testing under 6-fold cross-validation procedure (referred to as “All set” henceforth). Second, a set of genes that were deemed important for prediction consistently in each round, that is, in the discrimination of resistant strains from susceptible strains, were used along with phenotypes for training, validating, and testing the machine learning algorithms (“Intersection set”). Third, as many genes as in the intersection set were randomly sampled from All set and then used for training, validating, and testing of machine learning algorithms; performance averaged over ten such random replicates was assessed for each algorithm (“Random set”). This procedure is similar to that described in our previous study [12]. As the performance with Intersection set is comparable or better than with All set, we screened the Intersection set for genes that have not yet been implicated in AMR and subjected them to homology modeling followed by computational docking experiments involving the predicted structures of their protein products and those of the antibiotics that are neutralized by strains containing these genes. We focused on strains that are resistant to specific antibiotic(s) but lack genes known to confer resistance to those antibiotics.

### 2.7. Homology Modeling with MODELLER

The RSCB Protein Data Bank (PDB) database (https://www.rscb.org accessed on 15 May 2020) was downloaded, and a local BLAST database was created for homology modeling that entailed alignment of query protein sequences against the database sequences using BLAST. The BLAST “hits” (database sequences with significant similarity to query sequences) meeting the criteria of query coverage 70% and percent identity 30% were selected for homology modeling using MODELLER 10.1 (https://saslilab.org/modeller/ accessed on 15 May 2020). Command line align2d was invoked for template alignment based on dynamic programming algorithm of MODELLER, and of five 3D models constructed by AutoModel class; the best model with highest DOPE score was selected for docking for each target template, as described in the Modeller tutorial [17].

### 2.8. Receptor/Ligand Preparation and Docking with AutoDock Vina Smina

The AutoDock Vina v1.1.2 (https://vina.scripps.edu accessed on 15 May 2020) was downloaded and installed locally. The best PDB models for receptors (target proteins) were then prepped by removing water molecules to avoid interference for ligand docking. Further preparation was conducted by removing heteroatoms, repairing hydrogens, and finally adding Kollman/Gasteiger charges as described in AutoDock (https://autodock.scripps.edu/ accessed on 15 May 2020). The structured data file (SDF) of the respective antibiotics (ligands) were downloaded from PubChem (https://pubchem.ncbi.nlm.nih.gov accessed on 15 May 2020) and converted to PDB format file with OpenBabel v2.3.1 [18]. The PDB ligands were also similarly prepared via the command line AutoDock ligand preparation as described in [19]. Then the homology modeled receptors (target proteins) were docked with the respective ligands (antibiotics), and ligand-receptor binding free energy (ΔG, kcal/mol) was scored via the AutoDock Vina Smina fork for each docking experiment [20].

### 2.9. Method Validation

A redocking analysis to validate the method used was conducted by downloading 262 crystal structures of antimicrobial subclass carbapenem and 586 crystal structures of AMR protein class aminoglycoside from the PDB. The heteroatoms were filtered via BIOPython PDB Parser [21], and molecular docking was carried out with the proteins and the respective ligands (antibiotics). Of the top 5 simulations, the method’s validity was checked by calculating root-mean-square deviation lower bound (RMSD l.b.) and upper bound (RMSD u.b.) so as to establish the optimal binding model of the ligand with the receptor. Then the conformations of highest binding affinity and successful binding were delimited with a threshold of less than 2Å.

Similarly, a docking experiment was performed with random sampled ligands, with the goal to examine whether the overall binding affinity between the predicted receptors and the respective ligands is higher than that between the receptors and randomly sampled ligands. If this is so, it further validates the predictions. Post docking analyses, visualizations of the conformations were conducted using UCSF Chimera v1.15 [22], AutoDock Tools [23], and Discovery Studio Visualizer v21.1.0.20298 [24].

The machine learning framework and workflow is illustrated in Figure 1.

## 3. Results

### 3.1. Performance Assessment of the Machine Learning Framework

Assessment of machine learning algorithms in predicting resistance/susceptibility traits of bacterial strains in their response to specific antibiotics revealed that overall, Intersection set yielded optimal performance (assessed based on F1 score). This set contains genes that were deemed important in discrimination between susceptible and resistant strains by tree-based machine learning programs in each round of the cross-validation procedure. This is a very small number of genes compared to all genes in a genome that were used in All set analysis. One can speculate that using such a small fraction of genes from a strain may not render a robust classifier due to loss of information, however, our results show that the performance is comparable, often better than that from the usage of all genes, by most of the algorithms. This demonstrates that these handful of genes are indeed among the main players in conferring the traits and using all genes that may confound the discriminative capability of machine learning programs. Furthermore, as expected, the performance yielded by the Random set, with as many genes randomly sampled from All set as in the Intersection set, is almost always much lower than the performance attained using the Intersection set, further reinforcing the importance of Intersection set genes in predicting the traits and could thus be prioritized for further downstream analysis. Note that, in the Intersection set, a small number of already implicated AMR genes were among the important (top-ranked) features (genes); however, the proportion was small; only those important genes that were consistently found in all six rounds of the cross-validation were reported.

The overall accuracy of each machine learning algorithm in predicting the resistance trait, assessed in terms of F1 score (harmonic mean of recall and precision), is shown for each (species group, drug) combination in Figure 2, Figure 3, Figure 4 and Figure 5: (*Klebsiella pneumoniae*, Doripenem) in Figure 2a, (*Escherichia coli* and *Shigella*, Doripenem) in Figure 2b, (*Pseudomonas aeruginosa,* Imipenem) in Figure 2c, (*K. pneumoniae*, Meropenem) in Figure 3a, (*E. coli* and *Shigella*, Meropenem) in Figure 3b, (*Enterobacter*, Meropenem) in Figure 3c, (*K. pneumoniae*, Ertapenem) in Figure 4a, (*K. pneumoniae*, Imipenem) in Figure 4b, (*E. coli* and *Shigella*, Ertapenem) in Figure 4c, (*E. coli* and *Shigella*, Imipenem) in Figure 4d, (*P. aeruginosa,* Meropenem) in Figure 4e, (*Enterobacter*, Imipenem) in Figure 4f, (*Salmonella enterica,* Gentamicin) in Figure 5a, (*S. enterica,* Kanamycin) in Figure 5b, and (*S. enterica*, Streptomycin) in Figure 5c. Results show that the Intersection set yielded best overall performance. The overall accuracy generated by a best performing machine learning classifier was highest with Intersection set for all species group and drug combinations, except for (*E. coli* and *Shigella*, Imipenem) where the best performing DT classifier generated equivalent overall accuracy for All set and Intersection set and for (*S. enterica*, drug) where All set yielded the best performance (overall accuracy for both All set and Intersection set were over 90%). Performance assessment results for (species group, drug) combinations using other metrics, namely, precision, recall, AU ROC, AUPR, and Loo CV are summarized in the Appendix A. Assessment based on other metrics (AU ROC, AUPR) or variants of cross-validation (10-fold, Loo) showed similar trends, with Intersection set yielding best performance, with close to, or higher than 90% overall accuracy by optimal classifiers, in most instances.

### 3.2. Comparison with Our Previous Study Based on Only AMR Genes

In contrast to our previous study that only considered AMR genes, here we used all genes in a strain. The motivation here was to identify genes that have not yet been implicated in resistance to any antibiotics, while the motivation of our previous study was to identify AMR genes, already characterized for resistance, that may be responsible for resistance to an antibiotic in a strain lacking genes known to be involved in the resistance to the antibiotic. Because of different motivations of these studies, the current study could still hold its ground even if not rendering at par, or better, performance than attained in the previous study, because of its ability to discover genes not yet known to be involved in AMR that the previous approach can’t. Interestingly, the overall accuracy (F1 score) attained by the current approach is higher than that by the previous approach in most of the evaluations, further reinforcing confidence in novel resistance genes predicted by the current approach. The F1-scores by all the machine learning algorithms for the Intersection set or All set in this study was higher than the respective F1-scores rendered by the previous approach for (*K. pneumoniae*, Doripenem), (*K. pneumoniae*, Ertapenem), (*K. pneumoniae*, Imipenem), (*K. pneumoniae*, Meropenem) (Appendix A, F1-score for the former rose as high as 0.94 by RF and GBC); (*E. coli* and *Shigella*, Doripenem) (Appendix A; highest F1 of 0.92), (*E. coli* and *Shigella*, Imipenem) (Appendix A, highest F1 score of 0.97), and (*E. coli* and *Shigella*, Meropenem) (Appendix A, highest F1 score of 0.966); (*S. enterica*, Streptomycin) (Appendix A, highest F1-score of 0.97 by GBC), and (*S. enterica*, Kanamycin) (Appendix A, highest F1-score of 0.95 by ABC). For (*E. coli* and *Shigella*, Ertapenem), the F1 score by the previous approach (AMR only) was higher. For *C. jenuni* and *P. aeruginosa*, this comparison couldn’t be made as not all strains that have AST data have the complete genome data available in the NCBI pathogen database or the genotypic data were available but the respective AST data were missing. Note that the data for *C. jenuni* are no longer available in the database and the earlier available data for (*P. aeruginosa*, Doripemen) have now been replaced by new data for (*P. aeruginosa*, Imipemen) and (*P. aeruginosa*, Meropemen).

### 3.3. Novel AMR Genes

In each round of the 6-fold cross-validation, genes deemed important for discrimination were obtained. Genes common across all six gene sets were obtained for *K. pneumoniae* in (Appendix A), *E. coli* and *Shigella* (Appendix A), *P. aeruginosa* (Appendix A), *Enterobacter* (Appendix A), and *S. enterica* (Appendix A). Note the common set only includes both known AMR genes and novel AMR genes (not yet implicated in AMR). We focused on the latter (only on novel AMR genes) following verification of their presence in the resistant strains (note the common set may also contain genes that are absent in resistant strains but present in susceptible strains) and performed homology modeling and molecular docking analyses to garner support for novel predictions, as described in the following section. Lists of these novel AMR genes are provided in Table 2 for *K. pneumoniae* and in Appendix A for other species groups. Among top ranked novel genes in these lists, are several modifying enzymes with catalytic functions, such as, acetylation, phosphorylation, and adenylation that are known to initiate steric hinderance and thus decreases the affinity of the antimicrobials giving rise to AMR [25,26,27].

### 3.4. Binding Affinity of Proteins Encoded by Novel AMR Genes with Respective Antibiotics

The best models of proteins of interest, inferred using homology modeling, were used as receptors for respective antibiotics as ligands in molecular docking experiments. Energetically most stable ligand–receptor conformations displaying greatest affinity are provided in Table 3 for each species group considered in this study. We observed that proteins encoded by novel AMR genes identified in this study have high binding affinity with the respective antibiotics, comparable to that from simulated docking between known AMR proteins and respective antibiotics. In contrast, the overall binding affinities between randomly sampled proteins and respective antibiotics were significantly less (Appendix A for *K. pneumoniae*, *E. coli* and *Shigella*, *P. aeruginosa*, *Enterobacter*, and *S. enterica* respectively). This suggests that the proteins encoded by novel AMR genes form stable complexes with the respective antibiotics. Kanamycin and N-acetylneuraminate epimerase (WP_001643598) docked with the binding energy ΔG of −12.8 kcal/mol, displaying greatest affinity among all protein–drug combinations tested. The docking conformation for *K. pneumoniae* with receptor protein O-phosphotransferase (WP_000018330.1) and ligand Doripenem is shown in Figure 6, with interacting amino acids with hydrogen bond distance (Å), and hydrogen and hydrophobic interaction types indicated. Complete data generated from the docking experiments including details on interacting amino acids involved in hydrogen bonding, Affinity ΔG (Kcal/mol), RMSD lower bound, and RMSD upper bound are provided in Table 3. All other conformations displaying the mapped amino acids of the receptors docking with the respective ligands are shown in Appendix A for *K. pneumoniae*, *E. coli* and *Shigella*, *P. aeruginosa*, *Enterobacter*, and *S. enterica* respectively.

## 4. Discussion

Applications of machine learning in biology and medicine have gained momentum in recent years due to technological advances resulting in accumulation of mountains of biological or biomedical data. Machine learning has proved to be one of the most effective approaches in deciphering these data. In our studies of drug resistant bacterial pathogens, we leveraged the power of machine learning in deciphering genetic elements underlying resistance to various drugs. Previously, we focused on known AMR genomic loci and employed machine learning to uncover their yet unknown roles in conferring resistance to various antibiotics. This study focused on identifying novel AMR genes, that is, those that have not yet been implicated in resistance to any antibiotics. By taking an unbiased whole genome approach that allowed consideration of all protein-coding genes in a genome of interest, we demonstrated the power of machine learning in discovering new putative AMR genes, which was further supported by computational molecular docking experiments.

Whereas there are challenges abounding in identifying novel resistance elements in bacterial pathogens and machine learning is a step forward in right direction, these efforts could be confounded by intriguing mechanisms such as heteroresistance, where a subpopulation of strains might demonstrate variable phenotypes [28]. Switching of phenotype (resistant/susceptible) may occur with a change in environment, with genotype being invariant. Clearly, the gene–environment interaction also needs to be accounted for and perhaps, machine learning models, when presented with this new dimension of information, may learn new features to predict the phenotypes. In addition, non-protein-coding genes and regulatory sequences may also be significant players in imparting these differential phenotypes, and future machine learning based studies could focus on these elements as well, in addition to the protein-coding genes.

Since 1982, computational docking has been employed as an apparatus to predict protein–ligand interaction in CADD research; the lower the free energy change ΔG, the thermodynamically more stable is the target protein and ligand complex [14,29,30,31]. There are, however, challenges to address as molecular simulations ignore water molecules, which in a biological environment have substantial role in how biomolecules interact with each other [32]. For example, the strength of H-bonds between protein and ligand varies at neutral (pH 7), and molecular docking programs often ignore the polarity of strong and weak H-bonds that negates the orientation of water molecules in ligand and receptor interaction [33]. However, the main purpose of molecular docking is to assess interactions that yield most stable conformations, which of course need further validation in a wet lab setting. Note further, that not all resistance proteins could be interacting directly with antibiotics; although ML methods may identify such genes, they cannot be validated based on molecular docking. Functional genomics may aid in placing confidence over these predictions. One approach could be to assess their expression during resistance or use gene co-expression networks for understanding their role in resistance (see, for example, refs. [34,35] for such an approach that was used to characterize stress responsive genes). Novel resistance genes prioritized using computational approaches could be the candidates for experimental verification using wet lab assays, which may also provide insights into yet unknown mechanisms of resistance. We further emphasize that the machine learning approach presented here is applicable to not just pathogens, but could also be utilized for deciphering antimicrobial resistance in commensal microorganisms, which may serve as an AMR gene reservoir supplying AMR genes to pathogenic bacterial strains through horizontal gene transfer and thus rendering them resistant to drugs. Future studies can focus on these aspects as well.

## Figures and Tables

**Figure 2 microorganisms-10-02102-f002:**
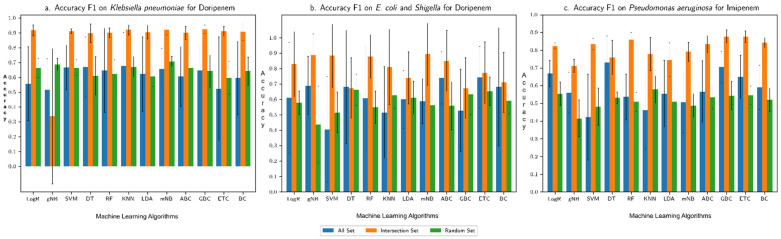
Overall accuracy (F1 score) produced by different machine learning algorithms in predicting resistance to (**a**) Doripenem for *K. pneumoniae*, (**b**) Doripenem for *E. coli* and *Shigella*, and (**c**) Imipenem for *P. aeruginosa*. “All Set” denotes all genes of a genome of interest, “Intersection Set” contains genes that consistently ranked high in their importance in discriminating resistant from susceptible strains in each round of cross-validation, and “Random Set” contains randomly sampled genes from All Set (same number of genes as in Intersection Set). Performance was assessed for each of three input datasets in a 6-fold cross-validation setting.

**Figure 3 microorganisms-10-02102-f003:**
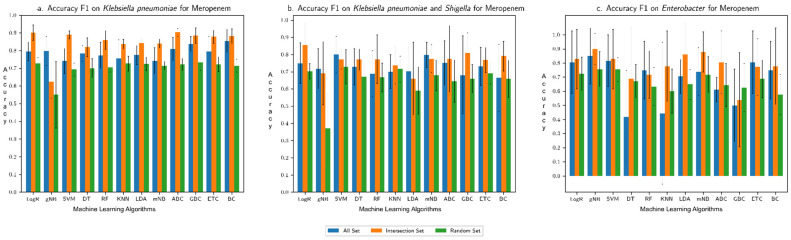
Overall accuracy (F1 score) produced by different machine learning algorithms in predicting resistance to Meropenem for (**a**) *K. pneumoniae*, (**b**) *E. coli* and *Shigella*, and (**c**) *Enterobacter*. “All Set” denotes all genes of a genome of interest, “Intersection Set” contains genes that consistently ranked high in their importance in discriminating resistant from susceptible strains in each round of cross-validation, and “Random Set” contains randomly sampled genes from All Set (same number of genes as in Intersection Set). Performance was assessed for each of three input datasets in a 6-fold cross-validation setting.

**Figure 4 microorganisms-10-02102-f004:**
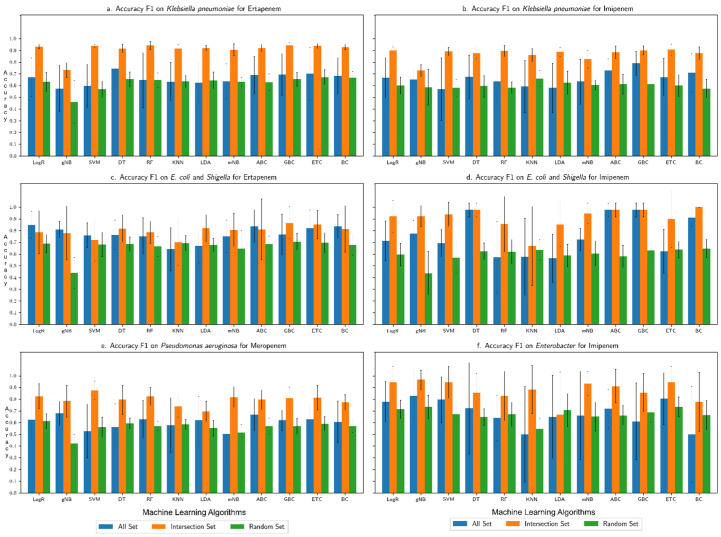
Overall accuracy (F1 score) produced by different machine learning algorithms in predicting resistance to (**a**) Ertapenem for *K. pneumoniae* (**b**) Imipenem for *K. pneumoniae*, (**c**) Ertapenem for *E. coli* and *Shigella*, (**d**) Imipenem for *E. coli* and *Shigella*, (**e**) Meropenem for *P. aeruginosa*, and (**f**) Imipenem for *Enterobacter*. “All Set” denotes all genes of a genome of interest, “Intersection Set” contains genes that consistently ranked high in their importance in discriminating resistant from susceptible strains in each round of cross-validation, and “Random Set” contains randomly sampled genes from All Set (same number of genes as in Intersection Set). Performance was assessed for each of three input datasets in a 6-fold cross-validation settings.

**Figure 5 microorganisms-10-02102-f005:**
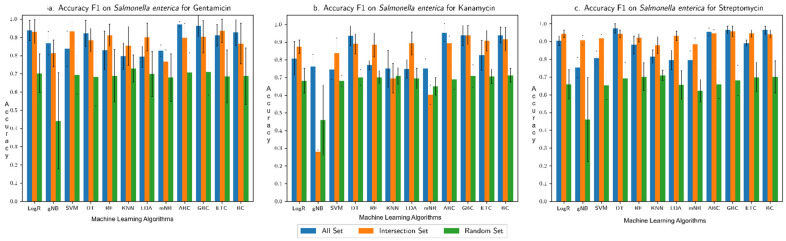
Overall accuracy (F1 score) produced by different machine learning algorithms in predicting resistance for *S. enterica* to (**a**) Gentamicin, (**b**) Kanamycin, and (**c**) Streptomycin. “All Set” denotes all genes of a genome of interest, “Intersection Set” contains genes that consistently ranked high in their importance in discriminating resistant from susceptible strains in each round of cross-validation, and “Random Set” contains randomly sampled genes from All Set (same number of genes as in Intersection Set). Performance was assessed for each of three input datasets in a 6-fold cross-validation setting.

**Figure 6 microorganisms-10-02102-f006:**
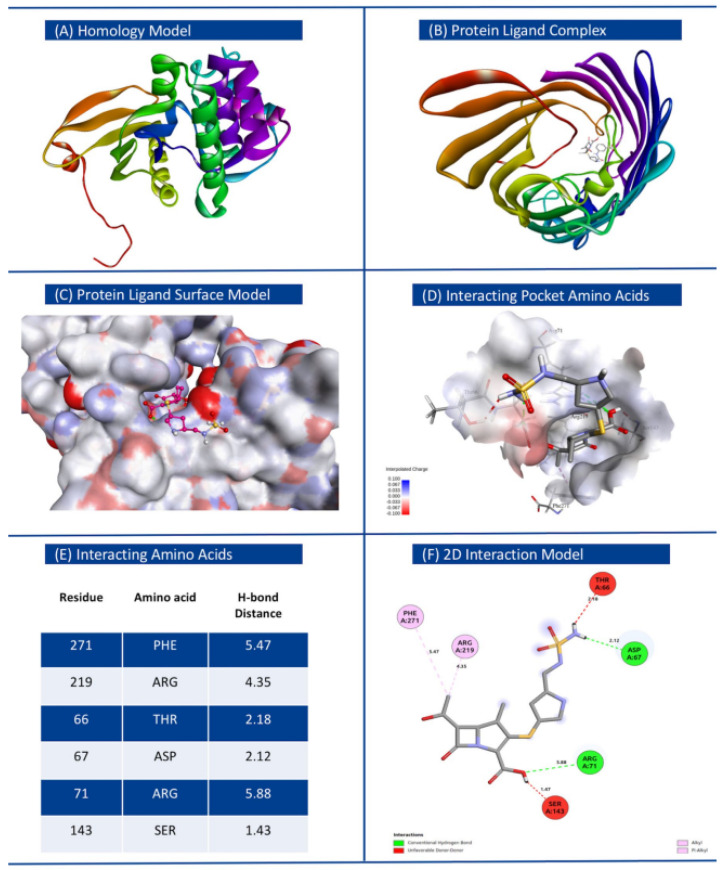
Homology model of protein O-phosphotransferase (WP_000018330.1) based on PDB template, and molecular docking with ligand (Doripenem) for *Klebsiella pneumoniae*. (**A**) Atomic resolution of target protein and its three-dimensional structure. (**B**) Protein ligand complex docked model (top view). (**C**) Binding surface plot of protein and ligand docking. (**D**) Interacting amino acids with interpolated charges inside the interacting pocket. (**E**) Details of residue, amino acids, and H-bond distance. (**F**) 2D interaction model with interaction bond types.

**Table 1 microorganisms-10-02102-t001:** Description of species groups with numbers of strains resistant and susceptible to antibiotics for each group. Also shown are total number of strains and total number of features (genes) for each species group and antibiotic combination, which were used in training, validation, and testing of different machine learning models in phenotype (resistance/susceptible) prediction.

SN	Bacteria	Antibiotics	Resistant	Susceptible	Total Strains	No. of Features (Genes)
1.	*Klebsiella pneumoniae*	Doripenem	19	19	38	23,611
		Ertapenem	45	45	90	23,611
		Imipenem	38	38	67	23,611
		Meropenem	69	69	138	23,611
2.	*E. coli* and *Shigella*	Doripenem	14	14	28	18,547
		Ertapenem	32	32	64	18,547
		Imipenem	22	22	44	18,547
		Meropenem	30	30	60	18,547
3.	*Pseudomonas aeruginosa*	Imipenem	41	41	82	12,945
		Meropenem	48	48	96	12,945
4.	*Enterobacter*	Imipenem	11	11	22	8987
		Meropenem	12	12	24	8987
5.	*Salmonella enterica*	Gentamicin	91	91	182	22,764
		Kanamycin	68	68	136	22,764
		Streptomycin	210	210	420	22,764

**Table 2 microorganisms-10-02102-t002:** Top ranked putative novel AMR genes in *K. pneumoniae*, not yet implicated as carbapenem resistant genes, following the verification of their presence in the resistant strains for each antibiotic (Doripenem, Ertapenem, Imipenem, and Meropenem). These novel AMR genes appeared among top ranked genes of importance by machine learning in each round of the 6-fold cross validation for *K. pneumoniae*.

SN.	Doripenem		Ertapenem		Imipenem		Meropenem	
	Accession	Protein Name/Description	Accession	Protein Name/Description	Accession	Protein Name/Description	Accession	Protein Name/Description
1	WP_004144294.1	AlpA family phage regulatory protein	WP_040215863.1	(2,3-dihydroxybenzoyl)adenylate synthase EntE	WP_004152650.1	DUF1833 family protein	WP_004152389.1	DUF2913 family protein
2	WP_048999684.1	Cu(+)/Ag(+) sensor histidine kinase	WP_016338366.1	DUF2913 family protein	WP_004218558.1	DUF2560 family protein	WP_071182194.1	DUF551 domain-containing protein
3	WP_000085883.1	DNA methylase	WP_000454193.1	DUF3330 domain-containing protein	WP_004152389.1	DUF2913 family protein	WP_004184301.1	DinB family protein
4	WP_004218558.1	DUF2560 family protein	WP_004171440.1	hypothetical protein	WP_062955115.1	GABA permease	WP_004199358.1	hypothetical protein
5	WP_032427884.1	hypothetical protein	WP_004151678.1	phosphodiester glycosidase family protein	WP_015065545.1	hypothetical protein	WP_004217321.1	IS6-like element IS26 family transposase
6	WP_023282973.1	arsenic transporter	WP_000219391.1	Mph(A) family macrolide 2’-phosphotransferase	WP_023300766.1	L-2-hydroxyglutarate oxidase	WP_071557814.1	IS6-like element IS6100 family transposase
7	WP_004144136.1	bifunctional phosphoribosyl-AMP cyclohydrolase/phosphoribosyl-ATP diphosphatase HisIE	WP_004141234.1	NAD(P)-binding domain-containing protein	WP_004199843.1	MetQ/NlpA family ABC transporter substrate-binding protein	WP_015367306.1	MAPEG family protein
8	WP_004152357.1	DNA polymerase III subunit theta	WP_139153029.1	Tn3 family transposase	WP_023290921.1	PD40 domain-containing protein	WP_032706418.1	PTS-dependent dihydroxyacetone kinase operon transcriptional regulator DhaR
9	WP_000997453.1	type VI secretion protein	WP_016240610.1	TraM recognition domain-containing protein	WP_068815603.1	PTS-dependent dihydroxyacetone kinase operon transcriptional regulator DhaR	WP_000166328.1	Rop family plasmid primer RNA-binding protein
10	WP_039817891.1	IS21 family transposase	WP_064162557.1	UDP-N-acetylmuramate dehydrogenase	WP_015703570.1	YqjK-like family protein	WP_019725280.1	SAM-dependent DNA methyltransferase
11	WP_000018330.1	aminoglycoside O-phosphotransferase APH(3’)-Ia	WP_004175424.1	YebG family protein	WP_004150845.1	dimethylsulfoxide reductase subunit B	WP_015365583.1	alanine transaminase AlaA
12	WP_004152392.1	IS3-like element Tn4401 family transposase	WP_002901627.1	YniB family protein	WP_009485658.1	glycerol dehydratase reactivase beta/small subunit family protein	WP_015365705.1	arsenic transporter
13	WP_004218009.1	glycosyl hydrolase family 2	WP_002888321.1	bis(5’-nucleosyl)-tetraphosphatase (symmetrical)	WP_020802362.1	multidrug/biocide efflux PACE transporter	WP_015704003.1	bifunctional biotin--[acetyl-CoA-carboxylase] ligase/biotin operon repressor BirA
14	WP_004198233.1	zinc-binding domain of primase-helicase family protein	WP_004152235.1	dimethylsulfoxide reductase subunit B	WP_000843500.1	DUF2933 domain-containing protein	WP_015703802.1	deacetylase
15	WP_001010162.1	arsenical pump-driving ATPase	WP_004191677.1	helix-hairpin-helix domain-containing protein	WP_009483845.1	SLC13/DASS family transporter	WP_000361389.1	plasmid partitioning/stability family protein

**Table 3 microorganisms-10-02102-t003:** List of novel AMR genes with their NCBI nomenclatures that were identified in this study. Each receptor–ligand interaction based on highest binding affinity is tabulated. Details regarding RMSD lower bound, RMSD upper bound, and respective interacting amino acids are also provided.

SN.	Bacteria	Ligand	Receptor	NCBI Nomenclature	Affinity, ΔG (Kcal/mol)	RMSD lb	RMSD ub	Interacting Amino Acids with the Ligand
1	*Klebsiella pneumoniae*	Doripenem	WP_000018330.1	MULTISPECIES: O-phosphotransferase	−7.9	0	0	ThrA66, Asp67, Arg71, Ser143, Arg219, Phe271
		Ertapenem	WP_049116479.1	MULTISPECIES: porin OmpK35	−10.6	0	0	Val16, Ala17, Trp133, Gly135, Ala347
		Imipenem	WP_064146913.1	PD40 domain-containing protein	−7.4	0	0	Tyr39, Leu88, Ala132, Pro197,
		Meropenem	WP_125961907.1	haloacid dehalogenase-like hydrolase	−8.9	0	0	Asp63, la111, Gly114, Ser254, gly256, ASer295, Gly296, Lys297,
2	*E. coli* and *Shigella*	Doripenem	WP_004201167.1	MULTISPECIES: bleomycin binding protein Ble-MBL	−7	0	0	Arg40, gly42, Gln44, Cys62, Arg65
		Ertapenem	WP001394742.1	MULTISPECIES: 6-phospho-beta-glucosidase BglA	−11.2	0	0	Gln20, His132, Asn177, Tyr315, Asn318, Phe333, Glu375, Trp423
		Imipenem	WP_004152394.1	MULTISPECIES: IS21-like element ISKpn7 family helper ATPase IstB	−7.2	0	0	Lys71, Asp77, Tyr76, Gly113, His116, Gly111, Lys114, Arg242,
		Meropenem	WP_032488579.1	MULTISPECIES: N-acetyltransferase	−8.7	0	0	Trp33, Ile99, Gln101, Pro138, Tyr149, Asp164
3	*Pseudomonas aeruginosa*	Imipenem	WP_031628187.1	autoinducer binding domain-containing protein	−7.7	0	0	Cys81, Asp103, Val112, gly115, Gln120,
		Meropenem	WP_023097121.1	MULTISPECIES: outer membrane protein OmpK	−8.9	0	0	His37, Glu57, Tyr72, Phe93, Lys248,, Asn277, Trp235
4	*Enterobacter*	Imipenem	WP_061096807.1	alpha, alpha-trehalase	−7.9	0	0	Glu171, Tyr173, trp175, Asn212, Gln223, Gly324, Ala321, Phe532, Trp534
		Meropenem	WP_023337592.1	MULTISPECIES: leucyl aminopeptidase family protein	−8.7	0	0	Trp488, Arg495, glu70, Arg161, Arg77, Gln110
5	*Salmonella enterica*	Gentamicin	WP_025766410.1	MULTISPECIES: class 1 integron integrase IntI1	−9.4	0	0	Arg114, Gln120, Glu185, Arg186, Tyr188,
		Kanamycin	WP_001643958.1	N-acetylneuraminate epimerase	−11.8	0	0	Ser89, Ser226, Ala282, Gly346, Ser350, Ser348,
		Streptomycin	WP_000465133.1	MULTISPECIES: thermonuclease family protein	−10.2	0	0	Asp36, Thr39, Ile40, Asp44, Asp91, Tyr93, Arg95, Tyr120

## Data Availability

Custom programs and the associated datasets have been made available at https://github.com/Janaksunuwar/AMRprediction_HomologyModeling_Docking, accessed on 24 August 2022.

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
