# Peer review of "Identification of Novel Antimicrobial Resistance Genes Using Machine Learning, Homology Modeling, and Molecular Docking"

_microorganisms, 2022, doi:10.3390/microorganisms10112102_

Round 1

Reviewer 1 Report

This manuscript describes the identification of novel antimicrobial resistance genes using machine learning feature selection methods as well as molecular docking approach. I personally find this manuscript favorable due to its ability of discovering uncharacterized AMR genes and validating the findings through docking. I only have a few questions that I hope the authors can address before the acceptance of the manuscript.

1. The authors used gene names as features and only implemented homology search on hypothetical proteins. Indeed their genes are downloaded from RefSeq and therefore should be standardized. However there may still be some gene names such as “[Multispecies] function” that may integrate the functions of multiple species into one, which makes the gene name complicated. Did the authors do anything to make sure that genes with very similar but may not be identical names were still put together?

2. Since the authors set a criteria of identity 30% and coverage 70% on the hypothetical protein homology search, I wonder whether the authors can validate their selection of criteria. I was also wondering whether the genes put together by their names also abide by the criteria.

3. The texts in Table 2 are too small to read.

4. How many known AMR genes were among the genes discovered in the intersection set? I understand that the authors mentioned that they focus on the unknown AMR genes in Discussion. I however am still wonder the proportion of known AMR genes that can be found by their intersection approach.

5. Did the authors implemented stratified cross validation?

6. Why both 10-fold and leave-one-out cross validation? Did the authors find something good in both parts? Or the two cross validation can complement each other?

7. Since the default number of trees is 10 for random forest as described by the authors, I guess they may have used an outdated scikit-learn package. The most current scikit-learn package revised the default random forest tree number to 100. This should not affect the result after all; however I would still suggest the authors update their software version and check out RF part again.

8. If possible please add software version number, for example for Vina and OpenBabel. Also the website for vina (line 208) does not work; it should be https://vina.scripps.edu/ instead of https://www.vina.scripps.edu/. The autodock webpage has the same problem.

9. Sharing of Jupyter Notebook on github does not abide by the code sharing practice of the computational biology community. If possible please share actual code and a readme indicating how the code can be executed.

Reviewer 2 Report

In the manuscript entitled “Identification of novel antimicrobial resistance genes using machine learning, homology modeling, and molecular docking”, the authors, Janak Sunuwar and Rajeev K. Azad, describe a novel strategy to identify antimicrobial resistance gene candidates among “hypothetical protein” encoding genes. The authors use machine learning and homology modeling to train and test their algorithm with pre-identified antimicrobial genes and later are able to assess the probability of candidate antimicrobial resistance genes in bacteria. This strategy and process could provide greatly needed renewed insight on anticipating antimicrobial resistance evolution among pathogenic and non-pathogenic bacteria. The topic is of high interest given the major worldwide public health concern with antimicrobial resistance infections. 

The manuscript is very well-written and organized. It is clear and concise. The introduction provides enough background, and demonstrates the need for the current study. The materials and methods are well described, although I am not an expert on ML. The results and discussion provide enough data, and demonstrate the potential of their strategy.

I have very few comments for the manuscript, which again, was very well-written and clear. 

  1. I believe the authors could make a point of the importance of expanding this type of analysis to commensal bacteria which are known to also carry antimicrobial resistance genes and being reservoirs of AMR genes for pathogens, through horizontal gene transfer

  2. L. 127-128: the authors provide a clear conventional threshold for the association of “hypothetical proteins” with specific nomenclatures. I believe it is important to state that while these hypothetical genes might become associated with specific nomenclatures there may be mis-association. This mis-association could be propagated by future readers unless the text warns the readers of this possibility. 

Minor points:

- L. 61: paragraph in the sentence.

- Fig 2-5: add x-axis and y-axis titles for each figure. It might not be as necessary to show so many figures. The authors might want to reconsider whether all are necessary.

- Fig 2-5: The authors should provide a broader explanation on the statistical error spread of the classifiers. It is unclear whether any of the classifiers demonstrates a significant difference between the Intersection set and the Random set.

- Statistical analysis of the results are not described in the Materials and Methods section.

Overall, I greatly enjoyed reading this manuscript. It has a broad audience potential, and is thought provoking.

Round 2

Reviewer 1 Report

I have only one comment: besides answering my questions/concerns, please also consider adding the response content into the manuscript. One such example is stratified cross-validation, in which the authors answered my question but told me that this can be seen in github code. This however is not enough, as not just me but other people will be interested in knowing whether you guys are using stratified cross-validation. Therefore it is better to add this information into the manuscript instead of asking everyone to read the code.

Author Response

Thank you for your suggestion. We have mentioned stratified cross-validation in section 2.3. "Training, validation, and testing". We have also mentioned regarding the known AMR genes in the Intersection set at the end of first paragraph of section 3.1. "Performance assessment of the machine learning framework". Several other suggestions are also reflected in the manuscript. We have also provided our consent to publish the peer-review report alongside manuscript, so together, these should provide all information including those reconciled between reviewers and us. Thank you again!